# Injectable Lipid Emulsion and Clinical Outcomes in Patients Exclusively Receiving Parenteral Nutrition in an ICU: A Retrospective Cohort Study Using a Japanese Medical Claims Database

**DOI:** 10.3390/nu15122797

**Published:** 2023-06-19

**Authors:** Hideto Yasuda, Yuri Horikoshi, Satoru Kamoshita, Akiyoshi Kuroda, Takashi Moriya

**Affiliations:** 1Emergency and Critical Care Medicine, Jichi Medical University Saitama Medical Center, 1-847 Amanuma-cho, Ohmiya-ku, Saitama 330-0834, Japan; tmoriya@jichi.ac.jp; 2Medical Affairs Department, Research and Development Center, Otsuka Pharmaceutical Factory, Inc., 2-9 Kandatsukasa-machi, Chiyoda-ku, Tokyo 101-0048, Japan; horikoshi.yuri@otsuka.jp (Y.H.);; 3Research and Development Center, Otsuka Pharmaceutical Factory, Inc., 1-1 Kandaogawa-machi, Chiyoda-ku, Tokyo 101-0052, Japan

**Keywords:** parenteral nutrition, injectable lipid emulsion, ICU patients, clinical outcomes, real-world data

## Abstract

Guidelines for the nutritional management of critically ill patients recommend the use of injectable lipid emulsion (ILE) as part of parenteral nutrition (PN). The ILE’s impact on outcomes remains unclear. Associations between prescribed ILE and in-hospital mortality, hospital readmission, and hospital length of stay (LOS) in critically ill patients in the intensive care unit (ICU) were investigated. Patients who were ≥18 years old in an ICU from January 2010 through June 2020, receiving mechanical ventilation, and fasting for >7 days, were selected from a Japanese medical claims database and divided, based on prescribed ILE during days from 4 to 7 of ICU admission, into 2 groups, no-lipid and with-lipid. Associations between the with-lipid group and in-hospital mortality, hospital readmission, and hospital LOS were evaluated relative to the no-lipid group. Regression analyses and the Cox proportional hazards model were used to calculate the odds ratios (OR) and regression coefficients, and hazard ratios (HR) were adjusted for patient characteristics and parenteral energy and amino acid doses. A total of 20,773 patients were evaluated. Adjusted OR and HR (95% confidence interval) for in-hospital mortality were 0.66 (0.62–0.71) and 0.68 (0.64–0.72), respectively, for the with-lipid group relative to the no-lipid group. No significant differences between the two groups were observed for hospital readmission or hospital LOS. The use of ILE for days 4 to 7 in PN prescribed for critically ill patients, who were in an ICU receiving mechanical ventilation and fasting for more than 7 days, was associated with a significant reduction in in-hospital mortality.

## 1. Introduction

Injectable lipid emulsion (ILE) is the only source of essential fatty acids available to patients who depend on parenteral nutrition (PN) because of an intolerance to oral food intake and enteral nutrition (EN) [1,2]. Guidelines for the nutritional management of critically ill patients recommend ILE to be used as part of PN [3,4]. However, it appears that a certain number of patients receive PN without ILE [5,6]. Many critically ill patients suffer from malnutrition, and a certain number of them depend on PN [5]. Yet, when these patients receive PN without ILE, they can develop an essential fatty acid deficiency, and they may suffer adverse clinical outcomes.

A meta-analysis of studies between 1980 and 1996, before the strategy of permissive underfeeding gained traction, involved critically ill patients and suggested that those receiving PN without ILE had lower complication rates and equivalent mortality rates compared to those receiving PN with ILE [7]. On the other hand, a single-center study published in 2021 of a selected group of critically ill surgical patients with hepatic disorder reported the highest mortality rates for those treated with PN without lipids [8]. In addition, propofol, an anesthetic agent containing lipid as a solvent [9], is often used during mechanical ventilation, but, to our knowledge, there have been no studies of the effects of ILE on clinical outcomes in critically ill patients that have taken propofol into consideration.

The effects of lipid supplementation (taking propofol into account) on the clinical outcomes of critically ill patients receiving PN still need to be clarified. The aim of this study was to investigate the associations between prescribed ILE (including lipid emulsion derived from propofol) and in-hospital mortality, hospital readmission, and hospital lengths of stay (LOS) in patients in an intensive care unit (ICU) who were mechanically ventilated and fasting, using a large-scale Japanese national medical claims database.

## 2. Materials and Methods

### 2.1. Data Source

The medical claims database used for this retrospective cohort study was provided by Medical Data Vision (MDV) Co., Ltd. (Tokyo, Japan). At the time that it was used (October 2020), the MDV database covered approximately 33 million patients who had been treated at 400 hospitals, representing about 23% of the acute care hospitals operating in Japan. Data extracted included hospital admission and discharge dates, patient characteristics upon admission (i.e., age, sex, height, weight, primary diagnosis, comorbidities, activities of daily living, and consciousness levels), medical treatments and drugs prescribed during hospitalization, and clinical outcomes at the time of discharge. The International Statistical Classification of Diseases and Related Health Problems, 10th revision (ICD-10), was used to ascertain diagnoses (Appendix A). Japan-specific codes were used to identify medical treatments (Appendix A). The database was also used to obtain information about the administration to patients of any oral or enteral nutritional intake and all intravenous solutions. The intravenous product names, compositions, and numbers were used to calculate the quantities of parenteral energy, amino acids, lipid, and carbohydrates prescribed.

### 2.2. Patient Population

Patients included in this study were aged 18 years or older, were admitted to an ICU between January 2010 and June 2020, were in an ICU for 4 days or longer, received mechanical ventilation on any of the first 3 days of ICU admission, survived and remained hospitalized longer than 7 days, and were fasting (i.e., receiving no oral or enteral nutrition) for longer than 7 days. For this study, the day of admission to an ICU was considered to be day 1. Patients were excluded from the study who had missing data or suspected input data errors for height, weight, or prescribed parenteral energy amounts. Input errors were defined as height < 100 cm or ≥200 cm, weight < 10 kg or ≥200 kg, and/or 0 kcal of prescribed energy between days 1 and 7.

### 2.3. Patient Group

To explore the association between ILE and clinical outcomes, patients were divided into 2 groups: the no-lipid group, who were not prescribed lipids during days 4 through 7, and the with-lipid group, who were prescribed lipids during days 4 through 7. As it is recommended that the dose from artificial nutrition be limited during the initial 3 days of ICU admission [3], the presence of prescription days 4 through 7 in the ICU was used in this study. In Japan, only soybean oil containing injectable lipid (SO ILE) is available as the lipid in PN. Propofol is dissolved in soybean oil or both soybean oil and medium-chain fatty acids.

### 2.4. Endpoints

The primary endpoint was in-hospital mortality, and the secondary endpoints were hospital readmission and LOS. Only those patients who were discharged and had survived were evaluated for secondary endpoints. The definition of hospital readmission was reentering the original hospital within 30 days of being discharged.

### 2.5. Data Extraction

Data were extracted or calculated from the database and categorized as follows: age (<60, 60 to 69, 70 to 79, 80 to 89, or ≥90 years); sex; body mass index (BMI) (<6, ≥16 to <18.5, ≥18.5 to <22.5, ≥22.5 to <25, or ≥25, according to the WHO classification [10]); admission hospital bed numbers (<200, ≥200 to <500, or ≥500); admission year (from 2010 to 2011, 2012 to 2013, 2014 to 2015, 2016 to 2017, 2018 to 2019, or 2020); primary admission diagnosis (from 10 disease categories according to ICD-10 codes (Appendix A)); severity of comorbidities (0, 1 to 2, or ≥3; according to Charlson Comorbidity Index [CCI] score in combination with the algorithm established by Quan et al. [11]); activities of daily living (0, 5 to 20, 25 to 40, 45 to 60, 65 to 95, or 100, assessed at hospital admission after being critically ill, using Barthel Index [BI] [12]); level of consciousness according to the Japan Coma Scale (JCS) [13] (i.e., JCS0 [alert]; JCS1, 1-digit code [not fully alert, but awake without any stimuli]; JCS2, 2-digit code [arousable with stimulation]; or JCS3, 3-digit code [unarousable]); and surgeries undertaken between days of hospitalization and ICU admissions, based on Japan-specific codes (Appendix A).

Data extracted also included drugs prescribed or medical treatments received from day 1 to day 7 of hospital admission: catecholamines (dobutamine, dopamine, epinephrine, and norepinephrine); transfusions (fresh frozen plasma, platelets, and/or red blood cells); albumin; renal replacement therapy; intra-aortic balloon pump; plasmapheresis; extracorporeal membrane oxygenation (ECMO); nutritional support team (NST) intervention; rehabilitation (for cardiac macrovascular disease, cerebrovascular disease, disuse syndrome, locomotor disease, and/or respiratory disease); and/or feeding therapy.

For the calculations of daily prescribed parenteral doses of energy, amino acids, lipid, and carbohydrates, the extracted information about the number of parenteral nutrition products that were prescribed was used. Daily prescribed parenteral doses of lipid included propofol. Daily prescribed parenteral doses of energy included carbohydrate solutions that were used for maintaining hydration, glucose-containing extracellular fluid expansion, drug preparation, and propofol administration. Moreover, the days without nutrition (oral intake, EN, or parenteral amino acids/lipid) from day 8 to the day of discharge or in-hospital death were calculated.

Mean prescribed doses of energy (kcal/kg/day), amino acids (g/kg/day), lipids (g/kg/day), and carbohydrates (g/kg/day) during days 1 through 7 were calculated. The median of the prescribed daily doses of energy, amino acids, lipids, and carbohydrates by the group were calculated and graphed to demonstrate trends over time. The ideal body weight (=22 × [height, m^2^]) was used when doses per body weight were determined in order to avoid underestimating target doses of nutrition that might be caused by the tendency toward low body weight relative to height in Japanese patients.

### 2.6. Statistical Analysis

Statistical analysis was conducted by an independent third party (A2 Healthcare Corporation, Tokyo, Japan) to exclude bias and ensure transparency. Summary statistics of categorical variables were shown using frequencies and percentages, and those of continuous variables were shown using medians, first quartiles (Q1), and third quartiles (Q3). When there was missing data for BI or JCS, those characteristics were listed as “not available” (NA). The multivariate logistic regression analysis, the univariate regression analysis, and the analyses with the Cox proportional hazard model were performed with adjustment for the factors as follows, from clinical viewpoints: age; sex; BMI; admission hospital beds; admission years; primary diagnosis; Charlson Comorbidity Index; Barthel Index; JCS; surgery type; prescription/treatment of catecholamine, transfusion, albumin, renal replacement therapy, intra-aortic balloon pump, plasmapheresis, ECMO, NST intervention, and rehabilitation; mean prescribed doses of energy from day 1 to 7; mean doses of amino acids from day 1 to 7.

Associations between the 2 groups and in-hospital mortality or hospital readmission were analyzed with univariate logistic regression. Associations were also analyzed with multivariate logistic regression analysis, with adjustments for the factors described above. Odds ratios (OR), adjusted odds ratios (AOR), and 95% confidence intervals (CI) were calculated for the with-lipid group, with the no-lipid group being used as the reference. 

Associations between the two groups and hospital LOS were analyzed with univariate regression. Associations were also analyzed with multiple regression, adjusting for the two lipid groups and the factors described above. Regression coefficients before and after adjustments and 95% CI were calculated for the with-lipid group, with the no-lipid group being used as the reference.

Kaplan–Meier survival curves were generated for in-hospital mortality, and the log-rank test was performed to investigate the differences between the two groups. Furthermore, the Cox proportional hazard model was used to calculate adjusted hazard ratios (AHR) and 95% CI for the with-lipid group, with the no-lipid group as the reference. The hazard ratios were adjusted for the factors above. Patients discharged alive were censored on the day of discharge, and inpatients surviving for 90 days or longer were censored on Day 90.

A sensitivity analysis was performed after dividing the with-lipid group into 3 lipid subgroups, based on whether ILE or propofol was prescribed during days 4 to 7, as follows: ILE-only subgroup; ILE+ propofol subgroup; and propofol-only subgroup. The no-lipid group was used as the reference in the analysis. Associations between the four subgroups, including the no-lipid group and in-hospital mortality or hospital readmission, were analyzed with univariate logistic regression. Moreover, these associations were analyzed with multivariate logistic regression, adjusting for the four subgroups and the factors described above. OR, AOR, and 95% CI were calculated for the three subgroups, with the no-lipid group used as the reference. Associations between the four subgroups, including the no-lipid group and hospital LOS, were analyzed with univariate regression. Furthermore, these associations were analyzed with multiple regression, adjusting for the four subgroups and the factors described above. Regression coefficients before and after adjustment and 95% CI were calculated for the three with-lipid subgroups, with the no-lipid group used as the reference.

All of the AORs, adjusted regression coefficients, and AHRs above were calculated using 2 models, Model 1 and Model 2. Model 1 did not include the days without nutrition from day 8 to discharge or in-hospital death as an adjustment factor, but Model 2 included this information. All statistical analyses were performed using SAS, version 9.4 (SAS Institute, Inc.; Cary, NC, USA). A two-sided significance level of 5% was used.

## 3. Results

### 3.1. Patient Characteristics

Of the 23,406 patients in an ICU receiving mechanical ventilation and fasting long-term (>7 days), 20,773 met the inclusion criteria (Figure 1). Of those, 12,312 (59.3%) patients were in the no-lipid group, and 8461 (40.7%) patients were in the with-lipid group (Table 1). A total of 4525 (36.8%) patients in the no-lipid group and 2309 (27.3%) patients in the with-lipid group were 80 years or older, while 2746 (22.3%) patients in the no-lipid group and 1466 (17.3%) patients in the with-lipid group had BMI at or below 18.5. The most prevalent primary diagnosis in the no-lipid group was a disease of the digestive system in 1777 (14.8%) patients, followed by cerebrovascular disorders in 1664 (13.5%) patients, and other circulatory system diseases in 1664 (13.5%) patients. The most prevalent primary diagnosis in the with-lipid group was other circulatory system diseases in 1603 (18.9%) patients, followed by the disease of the digestive system in 1324 (15.6%) patients, and neoplasm in 1064 (12.6%) patients. 

The median (Q1, Q3) of the mean prescribed daily energy doses during days 1 through 7 was 6.9 (3.8, 11.7) kcal/kg in the no-lipid group and 10.6 (6.8, 15.7) kcal/kg in the with-lipid group (Table 1). The median (Q1, Q3) of the mean of daily prescribed lipid doses during days 1 through 7 was 0.00 (0.00, 0.01) g/kg in the no-lipid group and 0.13 (0.07, 0.22) g/kg in the with-lipid group. The median (Q1, Q3) of the mean prescribed daily carbohydrate doses during days 1 through 7 was 1.5 (0.9, 2.5) g/kg in the no-lipid group and 2.0 (1.3, 3.1) g/kg in the with-lipid group. The medians of the prescribed energy, amino acid, lipid, and carbohydrate doses for each group were calculated for days 1 through 7 to illustrate the changes in the prescribed doses over time (Figure 2). On day 7, the median (Q1, Q3) of the prescribed energy dose was 8.8 (3.7, 16.7) kcal/kg/day for the no-lipid group and 14.6 (7.4, 21.4) kcal/kg/day for the with-lipid group, and the median (Q1, Q3) of the prescribed amino acids was 0.29 (0.00, 0.57) g/kg/day for the no-lipid group and 0.42 (0.00, 0.65) g/kg/day for the with-lipid group.

### 3.2. Endpoints

In-hospital mortality occurred in 5863 (47.6%) patients in the no-lipid group but in only 2958 (35.0%) patients in the with-lipid group (Table 2). The AORs (95% CI) of in-hospital mortality for the with-lipid group relative to the no-lipid group were 0.62 (0.58–0.67) (Model 1) and 0.66 (0.62–0.71) (Model 2). Kaplan–Meier curves were generated, and the survival rate for the with-lipid group was significantly higher than for the no-lipid group (*p* < 0.001) (Figure 3). The AHRs (95% CI) of in-hospital mortality for the with-lipid group relative to the no-lipid group were 0.68 (0.64–0.71) (Model 1) and 0.70 (0.67–0.73) (Model 2). No significant differences were found in the AORs for hospital readmission or the adjusted regression coefficients for hospital LOS between the two groups (Table 2).

### 3.3. Sensitivity Analysis

Among the patients in the with-lipid group, there were 1073 (12.7%) patients in the ILE-only subgroup, 541 (6.4%) patients in the ILE+ propofol subgroup, and 6847 (80.9%) patients in the propofol-only subgroup (Table 3). For Model 1, the AORs (95% CI) of in-hospital mortality relative to the no-lipid group were 0.82 (0.71–0.96) for the ILE-only group, 0.68 (0.55–0.84) for the ILE+propofol group, and 0.59 (0.55–0.64) for the propofol-only group. For Model 2, they were 0.87 (0.75–1.01) for the ILE-only group, 0.71 (0.57–0.87) for the ILE+propofol group, and 0.63 (0.58–0.68) for the propofol-only group. No significant differences were found in the AORs of hospital readmission or the adjusted regression coefficients of hospital LOS between each of the three with-lipid subgroups and the no-lipid group.

## 4. Discussion

We investigated the effects of ILE prescribed during days 4 through 7 of ICU admissions on the clinical outcomes of patients who were mechanically ventilated and fasting for more than 7 days. We found a significant inverse association between prescribed ILE and in-hospital mortality. No significant differences were observed for hospital readmission or hospital LOS. Our study involving more than 20,000 patients provides strong evidence that ILE prescribed in the period soon after admission to an ICU may reduce in-hospital mortality in critically ill patients.

An older meta-analysis that included 2211 critically ill patients receiving PN reported that patients without ILE had lower complication rates and equivalent mortality rates compared to those who received PN with ILE [7]. However, that report only included studies conducted before 1996, and many of the patients received high energy doses of 35 kcal/kg/day or more. It is possible that the benefits of adding ILE may have been masked by the effects of overfeeding, which can adversely impact prognosis in critically ill patients [14]. On the other hand, our study of 20,773 patients involved a larger sample size and lower prescribed energy doses (8.8 kcal/kg/day in the no-lipid group and 14.6 kcal/kg/day in the with-lipid group). It may be that the inclusion of ILE in PN for patients in ICU exerts a favorable impact on mortality only when overfeeding is avoided. 

Another reason for the different conclusions between the two studies might be complex differences in control groups or patient characteristics. Among the control groups, compared to the study group (PN with ILE group) in the trials reviewed for the meta-analysis [7], were 5% dextrose solution group, EN group, or oral feeding group, and the conclusion of the meta-analysis might not come from the influence of ILE alone. Almost all study patients for the trials reviewed for the meta-analysis [7] were postoperative patients, not limited to ICU patients. In addition, many patients in such trials received more lipids and amino acids than the study patients in our current study did.

In this study, the prescribed amino acid dose was 0.16 g/kg/d in the no-lipid group and 0.23 g/kg/d in the with-lipid group, both of which were lower than that in the guidelines [3,4] and study results in other countries, except Japan [15,16]. However, the low prescribed dose of amino acids shown in this study was similar to other previous surveys of clinical nutrition management in Japanese ICUs [5,17]. In Japan, prescribed energy tends to be reduced as increasing the awareness of permissive underfeeding, while the awareness of the importance of amino acids is low, and amino acid solutions are seldom added to the 2-in-1 products widely used in Japan, which seems to lead to the low dose of prescribed amino acid [5]. 

A possible reason for the favorable impact of ILE on hospital survival is its protein-sparing effect. One systematic review of critically ill patients receiving PN reported that free fatty acid concentrations in the blood were maintained; nitrogen balance was improved, and, most importantly, proteins were spared in groups receiving PN with ILE [1]. A study of critically ill patients suggested that another benefit of ILE as part of PN was the suppression of endogenous glucose production [18]. When endogenous glucose is produced, amino acids are consumed [19]. The findings of these studies, along with our results, suggest that the protein-sparing effect of ILE may be at least one of the explanations for the improvements in the survival of critically ill patients who receive ILE supplementation. In addition, excessive carbohydrate dosing can be avoided with the use of ILE, which may result in an avoidance of hyperglycemia [20]. Moreover, the avoidance of hyperglycemia may lead to the improvement of clinical outcomes. However, in this study, the median (Q1, Q3) of the carbohydrate dose for the no-lipid group of 1.5 (0.9, 2.5) g/kg/d was lower than that for the with-lipid group, 2.0 (1.3, 3.1) g/kg/d; therefore, it is unlikely that the no-lipid group had more patients with hyperglycemia. As the database used in this study does not include data on plasma glucose levels, we cannot discuss the issue at this moment. In the future, the effects of ILE on plasma glucose control should be investigated. 

In Japan, whereas ILE use for critically ill patients has been limited, propofol has been used more often, particularly for patients who have been mechanically ventilated [5,9]. In our study, 7388 (35.6%) patients were prescribed propofol (which contains ILE as a solvent) during days 4 through 7 of their ICU admission. Because of this, we opted to include in our study a sensitivity analysis, in which clinical outcomes were compared between the three with-lipid subgroups (ILE-only, ILE+propofol, and propofol-only) and the no-lipid subgroup. Both prescribed energy and amino acid doses tended to be higher in the ILE-only group and the ILE+propofol group than in the propofol-only group (Appendix A). This result might be related to the comparatively high awareness of nutritional management among the physicians who prescribe ILE. As the prescribed energy and amino acid doses differed between subgroups, a multivariate analysis was performed where adjustment was made for not only patient characteristics but also parenteral energy and amino acid doses for the statistical analysis of subgroups. Relative to the no-lipid group, the AOR of in-hospital mortality for each of the three subgroups was significantly lower, with the exception of this difference not being quite statistically significant for the ILE-only group when the OR was additionally adjusted for days without nutrition from day 8 to day of discharge or in-hospital mortality (Model 2). It is possible that the small size of the ILE-only group relative to the No-lipid group was the reason a significant difference was not found between those two subgroups for in-hospital mortality.

In our study, only 1614 (7.8%) patients (excluding those who had only propofol prescribed) had ILE prescribed during days 4 through 7 of ICU admission. The limited clinical use of ILE in Japan is likely related to the market conditions in the country. In Japan, three-in-one bag formulations, which contain carbohydrates, amino acids, and lipids, are rarely available, and the only available lipid comes in the form of SO ILE. This lipid has a high relative content of *n*-6 unsaturated fatty acids, which has been reported to increase inflammatory reactions in critically ill patients [1]. Thus, clinicians in Japan may be avoiding the use of SO ILE, especially in patients admitted to an ICU. However, the use of either SO ILE or Mixed-oil ILE (containing medium-chain triglycerides, olive oil, fish oil, or mixtures of oils) is recommended during the first week of ICU admission for critically ill patients in the nutritional guidelines of the American Society for Parenteral and Enteral Nutrition [4]. Furthermore, our study showed a significant association between SO ILE being prescribed during days 4 through 7 of an ICU admission and a decrease in in-hospital mortality. Thus, even if ILE contains soybean oil, strong consideration should be given by clinicians to using this as part of PN in critically ill patients.

This study has several limitations. First, the patient characteristics used were extracted from a medical claims database, which may have contained entry errors or had missing data. The primary diseases of patients were coded with ICD-10 retrospectively, whereas the characterization of diagnoses prospectively is usually preferred. On the other hand, CCI was also applied retrospectively as a measure of comorbidities, and it has been validated in Japan as an accurate and reliable approach to characterizing comorbidities [21]. Second, we were unable to include characteristics pertaining to acute physiology, chronic health conditions, and disease severity, such as the acute physiology and chronic health evaluation II score [22] or the sequential organ failure assessment score [23], because laboratory data were not included in the database. To work around this, we extracted information about in-hospital prescriptions, treatments, and surgery, and we included the CCI, BI, and JCS for most patients. Then, to help determine whether disease severity impacted our study endpoints, we adjusted our regression analyses using all of these characteristics. However, unknown or residual confounding factors may still exist. In addition, the nutrition doses might be overestimated because the remaining and discarded amount after administration to patients was not considered. Finally, our results may not be possible to generalize to patients in other countries or those receiving more robust parenteral nutritional management. There are several reasons for this. Because the only lipid that is clinically available in Japan is soybean-oil-based, and propofol contains soybean oil alone or in combination with medium-chain fatty acids, our study did not include any other lipids, such as olive or fish oil, which are available in other countries. In addition, the prescribed energy, amino acid, and lipid doses in our study were relatively low, particularly compared with dose recommendations in guidelines. Moreover, all of our patients were fasting for longer than 7 days, and a large proportion of our patients were elderly (over 30% were 80 years old or older) or had a low BMI (over 20% with a BMI of 18.5 or less). These types of baseline characteristics related to disease and nutritional conditions may be more common and/or worse in patients in Japanese ICUs than in critically ill patients involved in European and American studies.

## 5. Conclusions

The addition of ILE for days from 4 to 7 prescribed for critically ill patients who were in an ICU receiving mechanical ventilation and fasting for more than 7 days was associated with a significant reduction in in-hospital mortality. No significant differences were observed for hospital readmission or hospital LOS.

## Figures and Tables

**Figure 1 nutrients-15-02797-f001:**
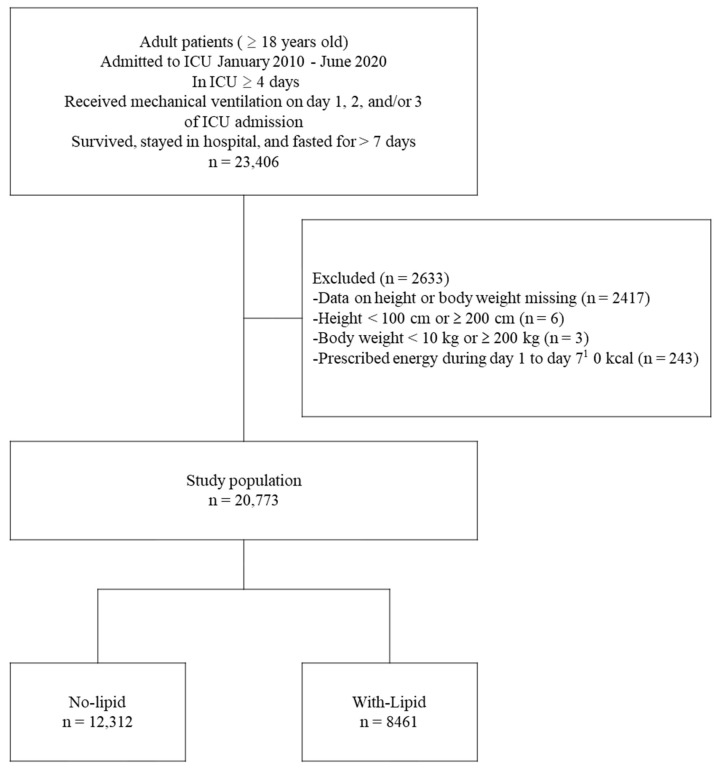
Study and patient disposition flowchart. ^1^ The day of admission to the ICU was considered day 1.

**Figure 2 nutrients-15-02797-f002:**
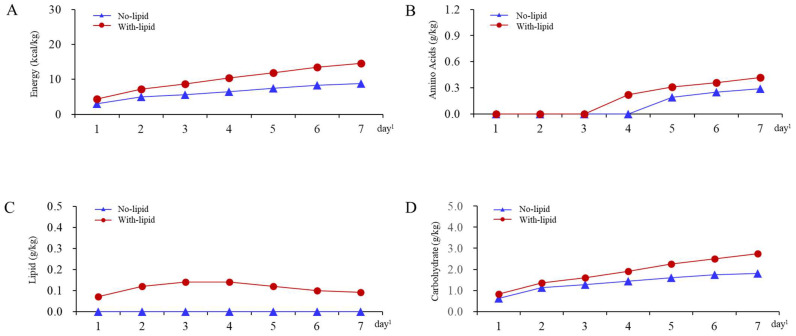
Median of prescribed daily doses of parenteral nutrition on days 1 through 7 of ICU admission in no-lipid and with-lipid groups, among 20,773 patients in Japan, from January 2010 to June 2020. The changes in the prescribed doses over time in an ICU are shown. The medians of the prescribed daily doses of energy and amino acids increased slightly for both groups between day 1 and day 7, whereas the medians of the doses of lipids decreased after day 2 in the with-lipid group. (**A**) The change in prescribed energy dose over time. (**B**) The change in prescribed amino acid dose over time. (**C**) The change in prescribed lipid dose over time. (**D**) The change in prescribed carbohydrate dose over time. ^1^ The day of admission to the ICU was considered day 1.

**Figure 3 nutrients-15-02797-f003:**
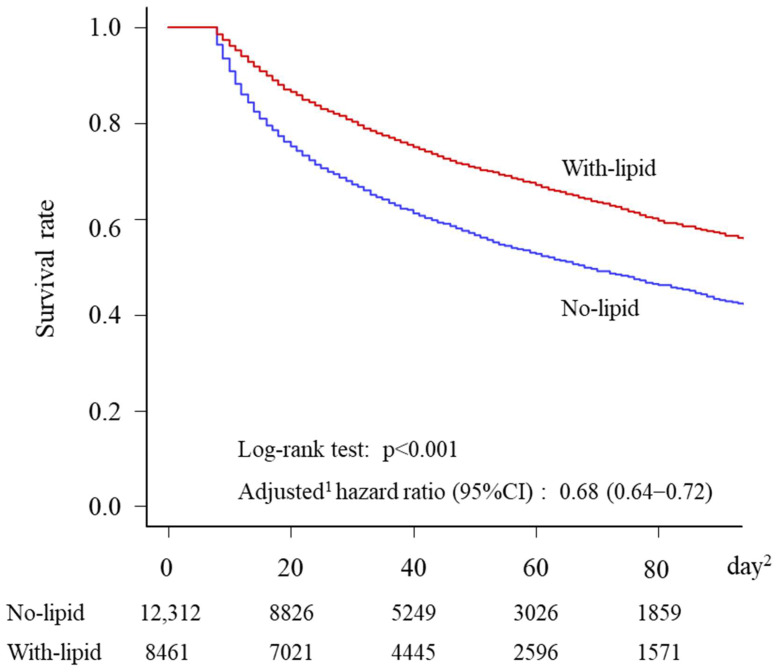
Kaplan–Meier survival curves and adjusted ^1^ hazard ratios of in-hospital mortality (days ^2^ 1 through 90) in no-lipid and with-lipid groups, among 20,773 patients in Japan, from January 2010 through June 2020. The survival rate for the with-lipid group was significantly higher than for the no-lipid group (*p* < 0.001). The adjusted hazard ratio of in-hospital mortality for the with-lipid group compared to the no-lipid group was 0.68. ^1^ Hazard ratios were adjusted for age, sex, BMI, admission hospital beds, primary diagnosis, year of admission, Charlson Comorbidity Index, Barthel Index, Japan Coma Scale, surgery, catecholamines, transfusion, albumin, renal replacement therapy, intra-aortic balloon pump, extracorporeal membrane oxygenation, nutritional support team intervention, rehabilitation, mean prescribed daily energy and amino acid doses from day 1 to day 7, and days without nutrition from day 8 to day of discharge or in-hospital death. ^2^ The day of admission to the ICU was considered day 1.

**Table 1 nutrients-15-02797-t001:** Characteristics of 20,773 patients in Japanese intensive care units, from January 2010 to June 2020, by groups based on prescribed injectable lipid emulsion during days ^1^ 4 through 7.

Characteristics	Categories	Patient Groups	
		No-Lipid *n* = 12,312	With-Lipid *n* = 8461
Age, years, *n* (%)	<60	1924 (15.6)	1713 (20.2)
	60–69	2286 (18.6)	1763 (20.8)
	70–79	3577 (29.1)	2676 (31.6)
	80–89	3712 (30.1)	2030 (24.0)
	≥90	813 (6.6)	279 (3.3)
Sex, *n* (%)	Male	7474 (60.7)	5678 (67.1)
	Female	4838 (39.3)	2783 (32.9)
Body mass index, kg/m^2^, *n* (%)	<16	880 (7.1)	439 (5.2)
	16–18.5	1866 (15.2)	1027 (12.1)
	18.5–22.5	4523 (36.7)	2955 (34.9)
	22.5–25	2386 (19.4)	1895 (22.4)
	≥25	2657 (21.6)	2145 (25.4)
Beds in admission hospital, *n* (%)	<200	323 (2.6)	362 (4.3)
	≥200, <500	7135 (58.0)	4741 (56.0)
	≥500	4854 (39.4)	3358 (39.7)
Admission year, *n* (%)	2010–2011	409 (3.3)	299 (3.5)
	2012–2013	1266 (10.3)	850 (10.0)
	2014–2015	2539 (20.6)	1770 (20.9)
	2016–2017	3554 (28.9)	2485 (29.4)
	2018–2019	3923 (31.9)	2642 (31.2)
	2020	621 (5.0)	415 (4.9)
Primary diagnosis ^2^, *n* (%)	Sepsis	681 (5.5)	570 (6.7)
	Neoplasm	1208 (9.8)	1064 (12.6)
	Diseases of the nervous system	555 (4.5)	215 (2.5)
	Ischemic heart disease	658 (5.3)	693 (8.2)
	Heart failure	800 (6.5)	340 (4.0)
	Cerebrovascular disorders	1664 (13.5)	747 (8.8)
	Other circulatory system diseases	1664 (13.5)	1603 (18.9)
	Pneumonia	549 (4.5)	257 (3.0)
	Interstitial respiratory diseases	365 (3.0)	233 (2.8)
	Other respiratory diseases	926 (7.5)	383 (4.5)
	Diseases of the digestive system	1777 (14.4)	1324 (15.6)
	Kidney diseases	162 (1.3)	109 (1.3)
	Injury, poisoning, other consequences external causes	551 (5.4)	395 (4.7)
	Other	752 (6.1)	528 (6.2)
Charlson Comorbidity Index, *n* (%)	0	6687 (54.3)	4298 (50.8)
	1–2	4072 (33.1)	2962 (35.0)
	≥3	1553 (12.6)	1201 (14.2)
Barthel Index, *n* (%)	100	2092 (17.0)	2131 (25.2)
	65–95	376 (3.1)	312 (3.7)
	45–60	381 (3.1)	268 (3.2)
	25–40	266 (2.2)	149 (1.8)
	5–20	564 (4.6)	342 (4.0)
	0	6840 (55.6)	4042 (47.8)
	NA	1793 (14.6)	1217 (14.4)
Japan Coma Scale, *n* (%)	0	5350 (43.5)	4680 (55.3)
	1–3	2123 (17.2)	1418 (16.8)
	10–30	1188 (9.6)	710 (8.4)
	100–300	3650 (29.6)	1653 (19.5)
	NA	1 (0.0)	0 (0.0)
Surgery ^3^, *n* (%)	Cardiovascular	765 (6.2)	1213 (14.3)
	Gastroenterological	2383 (19.4)	2109 (24.9)
	Cerebrovascular	923 (7.5)	437 (5.2)
	Respiratory	29 (0.2)	41 (0.5)
	Orthopedic	32 (0.3)	40 (0.5)
	Urological/Gynecological	63 (0.5)	41 (0.5)
	Multiple surgeries	41 (0.3)	62 (0.7)
	Other	172 (1.4)	194 (2.3)
	No surgery	7904 (64.2)	4324 (51.1)
Prescription/treatment ^4^, *n* (%)	Catecholamines	8530 (69.3)	6519 (77.0)
	Transfusion	5257 (42.7)	5129 (60.6)
	Albumin	4961 (40.3)	5032 (59.5)
	Renal replacement therapy	2319 (18.8)	2225 (26.3)
	Intra-aortic balloon pump	746 (6.1)	920 (10.9)
	Plasmapheresis	62 (0.5)	44 (0.5)
	ECMO	367 (3.0)	531 (6.3)
	Nutritional support team	233 (1.9)	173 (2.0)
	Rehabilitation ^5^	5075 (41.2)	3838 (45.4)
Parenteral nutrition ^6^, median (Q1, Q3)	Energy, kcal/kg/d	6.9 (3.8, 11.7)	10.6 (6.8, 15.7)
	Amino acids, g/kg/d	0.16 (0.00, 0.38)	0.23 (0.04, 0.43)
	Lipid, g/kg/d	0.00 (0.00, 0.01)	0.13 (0.07, 0.22)
	Carbohydrate, g/kg/d	1.5 (0.9, 2.5)	2.0 (1.3, 3.1)
Days without nutrition from day 8 ^7^, median (Q1, Q3)		0.0 (0.0, 1.0)	0.0 (0.0, 0.0)

Abbreviations: ECMO, extracorporeal membrane oxygenation; Q1, first quartile; Q3, third quartile. ^1^ The day of admission to the ICU was considered day 1. ^2^ Diagnoses based on International Statistical Classification of Diseases and Related Health Problems, 10th revision. ^3^ Between days of hospital and intensive care unit admissions, identified using Japan-specific codes. ^4^ Between days 1 and 7 of intensive care unit admission. ^5^ Feeding therapy and/or rehabilitation for cardiac macrovascular, cerebrovascular, disuse syndrome, locomotor, and/or respiratory diseases. ^6^ Medians of mean prescribed daily doses during days from 1 to 7 of intensive care unit admission. ^7^ Days without nutrition (oral intake, enteral nutrition, or parenteral amino acids/lipid) from day 8 to day of discharge or in-hospital death.

**Table 2 nutrients-15-02797-t002:** Clinical outcomes in 20,773 patients in Japanese intensive care units, from January 2010 through June 2020, by groups based on injectable lipid emulsion during days ^1^ 4 through 7.

	Patient Group		Odds Ratio/Regression Coefficient (95% CI) ^2^		
	No-Lipid *n* = 12,312	With-Lipid *n* = 8461	Unadjusted	Model 1 ^3^	Model 2 ^4^
In-hospital mortality, *n* (%)	5863 (47.6)	2958 (35.0)	0.59 (0.56–0.63)	0.62 (0.58–0.67)	0.66 (0.62–0.71)
Hospital readmission ^5,6^, *n* (%)	270 (4.2)	241 (4.4)	1.03 (0.85–1.25)	0.96 (0.78–1.18)	0.94 (0.76–1.15)
Hospital LOS ^5^, median (Q1, Q3)	45 (29, 71)	48 (31, 73)	1.96 (−0.43–4.35)	0.58 (−1.88–3.04)	0.78 (−1.69–3.25)

Abbreviations: CI, confidence interval; LOS, length of stay; Q1, first quartile; Q3, third quartile. ^1^ The day of admission to the ICU was considered day 1. ^2^ Odds ratios and regression coefficients calculated for with-lipid group, with no-lipid group used as reference. ^3^ Adjustments made for age, sex, BMI, beds in admission hospital, primary diagnosis, admission year, Charlson Comorbidity Index, Barthel Index, Japan Coma Scale, surgery, catecholamines, transfusion, albumin, renal replacement therapy, intra-aortic balloon pump, extracorporeal membrane oxygenation, nutritional support team, rehabilitation, and mean of prescribed daily doses of energy and amino acid on days 1 through 7. ^4^ Adjustments made for all of the variables used in Model 1, as well as days without nutrition from day 8 to day of discharge or in-hospital death. ^5^ Surviving discharged patients *n* = 6449 in the no-lipid group; *n* = 5503 in the with-lipid group. ^6^ Patients, who were readmitted within 30 days of discharge to the hospital of original admission.

**Table 3 nutrients-15-02797-t003:** Clinical outcomes in 20,773 patients in intensive care units in Japan, from January 2010 through June 2020, by subgroups based on whether ILE or propofol was prescribed on days ^1^ 4 through 7.

		Patient Subgroups	
	ILE-only	ILE+ propofol	Propofol-only
	*n* = 1073	*n* = 541	*n* = 6847
	Unadjusted odds ratio/regression coefficient (95% CI) ^2^		
In-hospital mortality	0.74 (0.64–0.85)	0.63 (0.52–0.77)	0.57 (0.54–0.61)
Hospital readmission ^3,4^	1.08 (0.71–1.65)	1.01 (0.56–1.82)	1.02 (0.84–1.26)
Hospital LOS ^5^, median	2.34 (−3.02–7.70)	5.24 (−2.08–12.55)	1.67 (−0.86–4.19)
	Model 1 ^5^ odds ratio/regression coefficient (95% CI) ^2^		
In-hospital mortality	0.82 (0.71–0.96)	0.68 (0.55–0.84)	0.59 (0.55–0.64)
Hospital readmission ^3,4^	1.01 (0.65–1.56)	0.93 (0.50–1.71)	0.96 (0.77–1.19)
Hospital LOS ^3^	2.39 (−2.91–7.70)	2.97 (−4.27–10.21)	0.10 (−2.52–2.71)
	Model 2 ^6^ odds ratio/regression coefficient (95% CI) ^2^		
In-hospital mortality	0.87 (0.75–1.01)	0.71 (0.57–0.87)	0.63 (0.58–0.68)
Hospital readmission ^3,4^	0.98 (0.64–1.52)	0.92 (0.50–1.70)	0.93 (0.75–1.16)
Hospital LOS ^3^	2.65 (−2.67–7.96)	3.18 (−4.07–10.42)	0.29 (−2.33–2.91)

Abbreviations: ILE, injectable lipid emulsion; CI, confidence interval; LOS, length of stay. ^1^ The day of admission to the ICU was considered day 1. ^2^ Odds ratios and regression coefficients calculated for with-lipid subgroups, with the no-lipid group (*n* = 12,312) used as reference. ^3^ Surviving discharged patients: *n* = 6449 in the no-lipid group, *n* = 650 in the ILE-only group, *n* = 346 in the ILE+propofol group, and *n* = 4507 in the propofol-only group. ^4^ Patients, who were readmitted within 30 days of discharge to the hospital of original admission. ^5^ Adjusted for age, sex, BMI, admission hospital beds, primary diagnosis, year of admission, Charlson Comorbidity Index, Barthel Index, Japan Coma Scale, surgery, catecholamines, transfusion, albumin, renal replacement therapy, intra-aortic balloon pump, extracorporeal membrane oxygenation, nutritional support team intervention, rehabilitation, and mean prescribed daily doses energy and amino acid from day 1 to day 7. ^6^ Adjusted for all of the variables used in Model 1, as well as days without nutrition from day 8 to day of discharge or in-hospital death.

## Data Availability

The data supporting the findings of this study are available from Otsuka Pharmaceutical Factory, Inc. Some restrictions apply to the availability of these data, which were used under license for the current study, and so are not publicly available. Data are available from the authors upon reasonable request and with the permission of Otsuka Pharmaceutical Factory, Inc.

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
