# Peer review of "Injectable Lipid Emulsion and Clinical Outcomes in Patients Exclusively Receiving Parenteral Nutrition in an ICU: A Retrospective Cohort Study Using a Japanese Medical Claims Database"

_nutrients, 2023, doi:10.3390/nu15122797_

Round 1

Reviewer 1 Report

The manuscript submitted by Yasuda et al. addresses the very interesting topic of clinical outcomes of the administration of injectable lipid emulsion in parenteral nutrition for ICU patients in Japan. The study design is well planned by the investigators and the obtained results are clearly presented and discussed. Here are a couple of comments that may help improve the manuscript before publication.

1.  Line 15-16. Please clarify. The use of ILE in Europe and the United States is a common practice so I can not agree with the statement that the ILE use is still limited.

2.  Line 23. Typo error: proportional

3.  Line 37. Typo error: nutritional

4.  It would be very interesting to present and discuss the energy, amino acids, and lipids provision in the subgroups (ILE-only, ILE+propofol, and Propofol-only) in the sensitivity analysis.

5.  Conclusion. The summary could be extended to include conclusions concerning hospital readmission and length of stay.

Author Response

Dear Reviewer #1,

We would like to express our sincere thanks to you for the thorough review of our manuscript and for the opportunity to submit a revised and improved version. We have carefully reviewed the comments and revised the manuscript on the basis of your comments. Our point-by-point responses to your comments are listed below this letter. Added or revised parts in the manuscript are written in red letters and removed parts in the manuscript are shown with a strike-out line.

Please note that we made a lot of revisions other than those according to your opinions to work on the reduction of the reception rate according to the journal instruction. Almost all of them are written in red letters or shown with a strike-out line.

All revised parts can be confirmed in the re-submission manuscript with track-change.

We hope that you find the current version of the manuscript suitable for publication. We will certainly be willing to make additional changes should they be required. Thank you for your consideration. We look forward to the publication of our manuscript in the Nutrients.

Sincerely,

Hideto Yasuda

Comments and Suggestions for Authors

  1. Line 15-16. Please clarify. The use of ILE in Europe and the United States is a common practice so I can not agree with the statement that the ILE use is still limited.

<Response>

Thank you for your comments and suggestions.

We revise the lines 15-16 as follows.

The ILE’s use is still limited and its impact on outcomes remains unclear.

  1. Line 23. Typo error: proportional.

<Response>

Thank you for your comments. We correct the typo error.

  1. Line 37. Typo error: nutritional.

<Response>

Thank you for your comments. We correct the typo error.

  1. It would be very interesting to present and discuss the energy, amino acids, and lipids provision in the subgroups (ILE-only, ILE+propofol, and Propofol-only) in the sensitivity analysis.

<Response>

Thank you for your comments and suggestions.

We prepared Supplementary Table 3 which shows patient characteristics including parenteral energy, amino acid, and lipid doses of the 3 subgroups. Also, the following sentences were added to lines 402-408. 

Both prescribed energy and amino acid doses tended to be higher in the ILE-only group and the ILE+propofol group than in the propofol-only group (Supplementary Table 3). This result might be related to the comparatively high awareness of nutritional management in the physicians who prescribe ILE. As the prescribed energy and amino acid doses differed between subgroups, a multivariate analysis was performed where adjustment was made for not only patient characteristics but also parenteral energy and amino acid doses for the statistical analysis of subgroups.

  1. Conclusion. The summary could be extended to include conclusions concerning hospital readmission and length of stay.

<Response>

Thank you for your comments and suggestions.

We add the following sentence in lines 348-34- and lines 461-462.

No significant differences were observed for hospital readmission or hospital LOS.

Reviewer 2 Report

The manuscript discusses the use of injectable lipid emulsion (ILE) in critically ill patients who received mechanical ventilation and fasted for over 7 days.  They were divided into two groups: those prescribed ILE and those without ILE as part of parenteral nutrition. However, the impact of ILE on patient outcomes remains uncertain but it was seen that the group receiving ILE had a significantly lower in-hospital mortality rate compared to the non-ILE group. There were no significant differences observed between the two groups in terms of hospital readmission or length of stay.

The authors paid attention to concomitant dextrose infusion from medicinal infusions and also the triglyceride content of propofol formulations which is a novel and commendable contribution! It is also interesting to see a Japanese manuscript on PN and the 20k ICU patient cohort.

The manuscript contributes with new knowledge . 

I have some suggestions for improvements:

*In eg table 1 - figures of aa in g/kg/d are given (0.16 vs 0.23). These figures are low when comparing with reference [3] and not really discussed

*If you looked also at carbohydrate intake from medicinal infusions - what were their numbers g/kg/d of dextrose? Can you share that and also give lipids in g/kg/d not just g/day.

*Was data from prescribed or administered doses?

*Figure 3 legend has some typographical issues with the numbered foot notes.

*In the discussion around row 310 you discuss a very complex material where there could be many factors not mentioned, energy differences, fatty acid differences, carbohydrate differences?  

*In the discussion - Especially the carbohydrate part and plasma glucose / insulin treatments are also necessary to discuss. Insulin is not mentioned in the manuscript but data are likely available in the database.

I have some suggestions for improvements:

* There are a lot of hyphens in the text that seem to be remnants of an earlier formatting, eg "retro-spective" and "data-base" in the title and eg in introduction row 37

*the English language is good but throughout the ICU is mentioned in a peculiar way. I would recommend to say "...in an ICU:" in the title or not using abbreviations at all in the title. Also in eg tables it says "in the intensive care unit in Japan" when it should say "in Japanese intensive care units" or equivalent since there are more than one ICU in Japan. 

Author Response

Dear Reviewer #2,

We would like to express our sincere thanks to you for the thorough review of our manuscript and for the opportunity to submit a revised and improved version. We have carefully reviewed the comments and revised the manuscript on the basis of your comments. Our point-by-point responses to your comments are listed below this letter. Added or revised parts in the manuscript are written in red letters and removed parts in the manuscript are shown with a strike-out line.

All revised parts can be confirmed in the re-submission manuscript with track-change.

Please note that we made a lot of revisions other than those according to your opinions to work on the reduction of the reception rate according to the journal instruction. Almost all of them are written in red letters or shown with a strike-out line.

We hope that you find the current version of the manuscript suitable for publication. We will certainly be willing to make additional changes should they be required. Thank you for your consideration. We look forward to the publication of our manuscript in the Nutrients.

Sincerely,

Hideto Yasuda

Comments and Suggestions for Authors

  1. In eg table 1 - figures of aa in g/kg/d are given (0.16 vs 0.23). These figures are low when comparing with reference [3] and not really discussed.

<Response>

Thank you for your comments and suggestions. As the prescribed parenteral nutrition doses including amino acids are low in this study, the study results might not be generalized in the US and European countries where active nutritional management is common. We think such issue is important; therefore, we described as follows in lines 445-452.

Finally, our results may not be possible to generalize to patients in other countries or those receiving more robust parenteral nutritional management. … In addition, the prescribed energy, amino acid, and lipid doses in our study were relatively low, particularly relative to the compared with dose recommended recommendations in guidelines

However, we did not address the reasons why the prescribed nutritional doses are low; therefore, we add the following paragraph in lines 370-378 to discuss the issues.

In this study, the prescribed amino acid dose was 0.16 g/kg/d in the No-lipid group and 0.23 g/kg/d in the With-lipid group, both of which were lower than that in the guidelines [3, 4] and study results in other countries except Japan [16, 17]. But the low prescribed dose of amino acids shown in this study was similar to other previous surveys of clinical nutrition management in Japanese ICUs [5, 18]. In Japan, prescribed energy is tended to be reduced as increasing the awareness of permissive underfeeding; while, the awareness of the importance of amino acids is low and amino acid solutions are seldom added to the 2-in-1 products widely used in Japan, which seems to lead the small prescribed amino acid dose [5].

  1. If you looked also at carbohydrate intake from medicinal infusions - what were their numbers g/kg/d of dextrose? Can you share that and also give lipids in g/kg/d not just g/day.

<Response>

Thank you for your comments and suggestions. As your suggestions, we add the carbohydrate data in Table 1 and Fig. 2. In addition, we revise the lipid data to that using the unit of g/kg/d.

  1. Was data from prescribed or administered doses?

<Response>

Thank you for your inquiry. We show the prescribed doses, not administered doses; therefore, we use the word of “prescribed” to avoid misreading. In addition, we address as follows in in lines 174-176.

For the calculations of daily prescribed parenteral doses of energy, amino acids, and lipid, were calculated using the extracted data of prescribed information about number of parenteral nutrition products that were prescribed were used.

A medical claims database was used in this study and the doses were calculated from the number of IV products recorded in the database. As the remained and discarded amount after administration to patients was not considered, which is thought to be one of the study limitations. Accordingly, we add the following sentence in lines 443-445.

In addition, the nutrition doses might be overestimated because the remained and discarded amount after administration to patients was not considered.

  1. Figure 3 legend has some typographical issues with the numbered foot notes.

<Response>

Thank you for your comments and suggestions. According to your comment, we write the number in foot notes in superscript.

  1. In the discussion around row 310 you discuss a very complex material where there could be many factors not mentioned, energy differences, fatty acid differences, carbohydrate differences?

<Response>

Thank you for your comments and suggestions. We have confirmed the papers referred to the meta-analysis of the Reference #6, and add the following sentences in lines 362--369 to discuss further differences to our study.

Another reason for the different conclusions between two studies might be complex differences in control groups or patient characteristics. Many of the control group, compared to the study group (PN with ILE group) in the trials reviewed for the meta-analysis was 5% dextrose solution group, EN group, or oral feeding group, and the conclusion of the meta-analysis might not come from the influence of ILE alone. Almost all study patients for the trials reviewed for the meta-analysis were postoperative patients not limited to ICU patients. In addition, many patients in such trials received more lipids and amino acids than the study patients in our current study did.

  1. In the discussion - Especially the carbohydrate part and plasma glucose / insulin treatments are also necessary to discuss. Insulin is not mentioned in the manuscript but data are likely available in the database.

<Response>

Thank you for your comments and suggestions. The point you addressed is very important and we consider that it is one of the issues to be studied in the future. We add the following discussion in lines 388-395.

In addition, an excessive carbohydrate dosing can be avoided by the use of ILE, which may result in an avoidance of hyperglycemia [21]. And the avoidance of hyperglycemia may lead to the improvement of clinical outcomes. However, in this study, median (Q1, Q3) of the carbohydrate dose for the No-Lipid group of 1.5 (0.9, 2.5) g/kg/d was lower than that for the With-Lipid group, 2.0 (1.3, 3.1) g/kg/d; therefore, it is unlikely that the No-Lipid group had more patients with hyperglycemia. As the database used in this study does not include the data of plasma glucose level, we cannot discuss the issue well at this moment. In the future, effects of ILE on the plasma glucose control should be investigated.

As you pointed out, the database used in this study includes the data on prescribed insulin dose; however, it does not include the data on plasma glucose level. In general, good blood glucose control is associated with the reduced insulin administration. However, discussion of the plasma glucose control based on the data of prescribed insulin dose alone may lead to misunderstanding. One of the reasons is that the low awareness of the presence of hyperglycemia might result in lower insulin administration and the other is that the strict plasma glucose control may lead the higher insulin administration. The effects of ILE use on plasma glucose concentration and insulin dose should be studied in the future.

  1. There are a lot of hyphens in the text that seem to be remnants of an earlier formatting, eg "retro-spective" and "data-base" in the title and eg in introduction row 37.

<Response>

Thank you for your comments. We correct them.

  1. The English language is good but throughout the ICU is mentioned in a peculiar way. I would recommend to say "...in an ICU:" in the title or not using abbreviations at all in the title. Also in eg tables it says "in the intensive care unit in Japan" when it should say "in Japanese intensive care units" or equivalent since there are more than one ICU in Japan.

<Response>

Thank you for your comments. We correct them.
